# ALITA-G: SELF-EVOLVING GENERATIVE AGENT FOR AGENT GENERATION

## ABSTRACT

Large language models (LLMs) perform better when scaffolded into agents with memory, tools, and feedback. Beyond this, self-evolving agents have emerged, but current work largely limits adaptation to prompt rewriting or failure retries. Therefore, we present ALITA-G, a self-evolution framework that transforms a general-purpose agent into a domain expert by systematically generating, abstracting, and curating Model Context Protocol (MCP) tools. In this framework, a generalist agent executes a curated suite of target-domain tasks and synthesizes candidate MCPs from successful trajectories. These are then abstracted to parameterized primitives and consolidated into a *MCP Box*. At inference time, ALITA-G performs retrieval-augmented MCP selection with the help of each tool's descriptions and use cases, before executing an agent equipped with the MCP Executor. Across several benchmarks GAIA, PathVQA, and Humanity's Last Exam, ALITA-G attains strong gains while reducing computation costs. On GAIA validation, it achieves $83.03\%$ pass@1 and $89.09\%$ pass@3, establishing a new state-of-the-art result while reducing mean tokens per example by approximately 15% relative to a strong baseline agent. ALITA-G thus provides a principled pathway from generalist capability to reusable, domain-specific competence, improving both accuracy and efficiency on complex reasoning tasks.

## 1 INTRODUCTION

Large language models (LLMs) have demonstrated strong performance across a wide range of tasks [1; 2]. However, a standalone LLM is still often insufficient for complex real-world tasks, especially those that demand professional domain knowledge and difficult multi-step reasoning. To further enhance their problem-solving capability, recent work constructs agentic systems around LLMs that decompose tasks, orchestrate tools and data sources, and iterate via feedback [3; 4; 5]. Embedding an LLM within an agentic system mitigates the limitations of its parametric knowledge and, by leveraging external knowledge sources and tools, enables deep research ability, demonstrating remarkable capabilities in task decomposition, tool coordination, and adaptive reasoning across diverse domains [6; 7]. Beyond these abilities, a distinguishing property of advanced agent systems is their potential for self-evolution [2; 8]: by leveraging self-generated content and both internal and external feedback, they can bootstrap their capabilities and, with minimal explicit human intervention, evolve into increasingly capable agent systems.

Despite rapid progress in self-evolving agents, current systems still exhibit limitations that constrain their evolutionary potential and downstream performance. Evolution is often narrow in scope: agents iteratively polish performance in a single target task or a restricted domain without the capacity to lift a general-purpose agent into a domain expert across a set of related tasks [4; 9]. At the same time, evolution is typically shallow in mechanism: many methods tune only a limited subset of modules or tools [10; 11], or a or rely on error-repair heuristics [12], instead of performing task-conditioned, end-to-end adaptation of the whole architecture. End-to-end evolution is important sincereal tasks demand planning, decomposition, tool use, and memory to improve together rather than in isolation. Likewise, transforming a general agent into a domain expert across a task set improves transfer and sample efficiency within that domain, supports robust generalization to new but related tasks, and sustains long-horizon improvement.

To address these limitations, we define a new paradigm of self-evolution: transforming a general-purpose agent into a domain expert across a set of tasks through task-conditioned, end-to-end adaptation. Building on this paradigm, we introduce ALITA-G, a framework that enables such transformation and achieves substantially improved performance within the target domain. to deep expertise and strong performance on domain-specific tasks. Our method employs a multi-execution strategy, where a generalist agent repeatedly engages the task collection and systematically synthesizes diverse Model Context Protocol (MCP) [13] components to capture, generalize, and adapt behaviors across executions. Across iterations, we harvest high-quality MCPs from successful runs and subject them to abstraction and refinement to build domain-specific MCP repositories, referred to as *MCP Box*. These repositories serve as specialized toolkits that support retrieval-augmented tool selection at inference time, allowing agents to dynamically identify and invoke the most contextually relevant MCPs for novel tasks in their specialization domain. From a system-level perspective, ALITA-G integrates two central dimensions. It is evolving as it end-to-end transforms a general agent into a domain specialist, and it is generative as it instantiates task-specific specialists on demand. This dual capability improves both the efficiency of agent construction and the effectiveness of domain problem solving.

We conduct comprehensive experiments across diverse benchmarks, GAIA [14], PathVQA [15], and Humanity's Last Exam [16], to validate the effectiveness of our approach. The results demonstrate that ALITA-G generates high-performing domain-specialist agents across multiple domains: these specialists deliver strong in-domain performance while reducing computational overhead relative to a generalist agent. On the challenging GAIA benchmark, our method achieves $83.03\%$ pass@1 and $89.09\%$ pass@3 accuracy, establishing a new state-of-the-art performance. Detailed ablations and analyses confirmed the necessity of each component and the advantages of our key hyperparameter choices. Our contribution can be summarized in three dimensions:

- We present ALITA-G, a novel self-evolution framework that transforms generalist agents into domain specialists to achieve substantially improved performance within a specific domain.

- We are the first to couple MCP abstraction with MCP-level retrieval-augmented generation (RAG) in a single framework. This design distills task-specific MCPs into reusable primitives and retrieves them at inference, yielding consistent gains in accuracy while reducing compute and latency.

- Across diverse benchmarks, our method improves performance while reducing compute; on the GAIA validation set, it achieves $83.03\%$ pass@1 and $89.09\%$ pass@3 (new SOTA), scales with MCP Box richness, and ablations verify the contribution of each component.

## 2 METHODS

We introduce ALITA-G, a novel framework for automatic agent generation that constructs task-specific agents through systematic MCP box curation and retrieval-augmented tool selection. Our approach addresses the fundamental challenge of agent design automation by leveraging task-driven MCP generation and intelligent tool filtering mechanisms, overcoming the limitations of prior methods that are narrow in scope or shallow in mechanism.

### 2.1 PROBLEM FORMULATION

Given a collection of target tasks $\mathcal{T} = \{(x_i, y_i)\}_{i=1}^{N}$ where $x_i$ represents task specifications and $y_i$ denotes desired outcomes, our objective is to automatically synthesize a specialized agent $\pi_{\text{specialized}}$ capable of effectively handling tasks within the domain defined by $\mathcal{T}$.

Formally, we aim to construct:

$$\pi_{\text{specialized}} = \text{Alita-G}(\mathcal{T}, \pi_{\text{master}}), \tag{1}$$

where $\pi_{\text{master}}$ is a powerful general-purpose agent system, and the resulting specialized agent should satisfy:

$$\mathbb{E}_{(x,y)\sim\mathcal{D}_{\text{target}}}[\mathbb{I}\{\pi_{\text{specialized}}(x) = y\}] > \mathbb{E}_{(x,y)\sim\mathcal{D}_{\text{target}}}[\mathbb{I}\{\pi_{\text{base}}(x) = y\}], \tag{2}$$

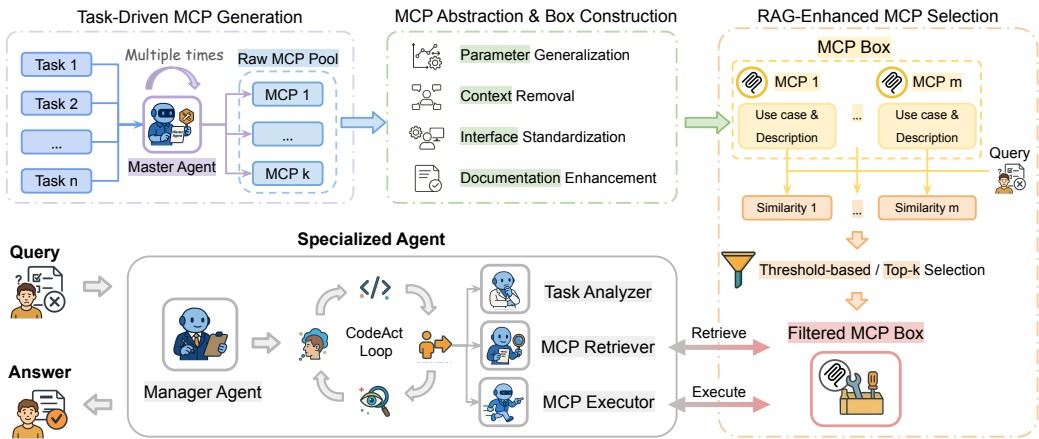

Figure 1: Overall workflow of ALITA-G. The process begins with task-driven MCP generation, where a Master Agent repeatedly executes target tasks and distills a pool of raw MCPs from successful trajectories. These MCPs are then abstracted and refined through parameter generalization, context removal, interface standardization, and documentation enhancement to form a reusable *MCP Box*. At inference time, the MCP Box supports RAG-enhanced tool selection: user queries are matched against MCP descriptions, and threshold/top-$k$ filtering yields a contextually relevant set of MCPs. Finally, a specialized agent—comprising a Manager Agent with a Task Analyzer, MCP Retriever, and MCP Executor—runs a CodeAct loop to retrieve and invoke the selected MCPs, thereby transforming a general-purpose agent into a domain specialist for end-to-end task solving.

where $\mathcal{D}_{\text{target}}$ represents the target task distribution and $\pi_{\text{base}}$ denotes a baseline agent without specialized capabilities.

## 2.2 TASK-DRIVEN MCP GENERATION

Our framework begins with systematic MCP generation through the master agent's task execution. When processing each task $(x_i, y_i) \in \mathcal{T}$, the master agent $\pi_{\text{master}}$ produces a reasoning trajectory:

$$\tau_i = (r_1^{(i)}, a_1^{(i)}, o_1^{(i)}, \ldots, r_{L_i}^{(i)}, a_{L_i}^{(i)}, o_{L_i}^{(i)}), \tag{3}$$

where $r_t^{(i)} \in \mathcal{R}$ represents reasoning tokens, $a_t^{(i)} \in \mathcal{A}$ denotes action tokens (including MCP generation calls), and $o_t^{(i)} \in \mathcal{O}$ corresponds to environmental observations.

During trajectory execution, the master agent is guided by explicit prompting to externalize reusable sub-solutions as self-contained MCPs rather than only producing final answers. The prompt instructs the agent to modularize complex sub-tasks into callable procedures with standardized interfaces and documentation, so that solving a task also expands the MCP pool for future reuse. We denote the $j$-th MCP generated during the execution of task $i$ as $\text{MCP}_{i,j}$, which includes both the executable code and associated metadata:

$$\text{MCP}_{i,j} = \{\text{code}_{i,j}, \text{description}_{i,j}, \text{use\_case}_{i,j}\}, \tag{4}$$

where $\text{description}_{i,j}$ provides a concise functional summary and $\text{use\_case}_{i,j}$ records the specific task context that triggered the MCP's creation.

To ensure quality and reliability, we implement a multi-execution strategy where each task $(x_i, y_i)$ is executed $K$ times, generating potentially different MCP variants. We collect MCPs only from successful executions where $\pi_{\text{master}}(x_i) = y_i$, forming the raw MCP pool:

$$\mathcal{L} = \{\text{MCP}_{i,j}^{(k)} \mid \pi_{\text{master}}^{(k)}(x_i) = y_i, \ i \in [N], j \in [J_{k,i}], k \in [K]\}, \tag{5}$$

where $J_{k,i}$ denotes the number of MCPs generated for task $i$ during the $k$-th execution run.

## 2.3 MCP Abstraction and Box Construction

Following the principles established in agent distillation literature, we apply abstraction techniques to transform instance-specific MCPs into generalizable tools. For each MCP in the raw pool $\mathcal{L}$, we employ a high-capacity language model to perform abstraction:

$$\widehat{\text{MCP}}_{i,j}^{(k)} = \text{LLM}_{\text{abstract}}(\text{MCP}_{i,j}^{(k)}) \tag{6}$$

The abstraction process accomplishes several critical transformations:

- **Parameter Generalization**: Replace hard-coded values with configurable parameters
- **Context Removal**: Eliminate task-specific references while preserving core functionality
- **Interface Standardization**: Ensure compatibility with FastMCP [17] protocol specifications, which is a high-performance implementation of the Model Context Protocol that provides optimized runtime support for dynamic tool integration and execution.
- **Documentation Enhancement**: Generate comprehensive docstrings and type annotations

Unlike traditional clustering approaches, our method preserves the diversity of MCP implementations to maximize coverage of potential task variations. The complete MCP box is defined as:

$$\mathcal{B} = \{\widehat{\text{MCP}}_m \mid m \in [M]\}, \tag{7}$$

where $M = |\mathcal{L}|$ represents the total number of abstracted MCPs, and each $\widehat{\text{MCP}}_m$ maintains its original metadata structure with abstracted code, preserved description, and use case information.

## 2.4 RAG-Enhanced MCP Selection

To address the challenge of tool relevance in diverse task scenarios, we introduce a retrieval-augmented generation mechanism for dynamic MCP selection. For each $\widehat{\text{MCP}}_m \in \mathcal{B}$, we construct a composite representation by concatenating its description and use case: $\text{context}_m = \text{description}_m \oplus \text{use\_case}_m$, where $\oplus$ denotes string concatenation.

Given a new task query $x_{\text{new}}$, we compute semantic embeddings for both the query and all MCP contexts using a pre-trained embedding model $\phi$:

$$\mathbf{e}_{\text{query}} = \phi(x_{\text{new}}), \mathbf{e}_m = \phi(\text{context}_m), \quad \forall m \in [M] \tag{8}$$

The relevance score between the query and each MCP is computed using cosine similarity:

$$s_m = \frac{\mathbf{e}_{\text{query}} \cdot \mathbf{e}_m}{\|\mathbf{e}_{\text{query}}\|_2 \|\mathbf{e}_m\|_2} \tag{9}$$

Our framework supports two complementary strategies for MCP selection based on the computed relevance scores:

**Threshold-based Selection**: We select MCPs whose relevance scores exceed a predefined threshold $\tau$:
$$\mathcal{B}_{\text{filtered}}^{\text{thresh}} = \{\widehat{\text{MCP}}_m \mid s_m \geq \tau, m \in [M]\} \tag{10}$$

This approach ensures that only sufficiently relevant tools are included, providing quality control over the selected MCP subset while maintaining flexibility in the number of selected tools.

**Top-k Selection**: Alternatively, we select the $k$ MCPs with the highest relevance scores:

$$\mathcal{B}_{\text{filtered}}^{\text{top-k}} = \{\widehat{\text{MCP}}_m \mid m \in \text{argsort}(\{s_j\}_{j=1}^M)[-k:]\} \tag{11}$$

This strategy guarantees a fixed number of tools for consistent computational overhead while ensuring that the most relevant MCPs are always selected, regardless of their absolute similarity scores.

The choice between threshold-based and top-k selection depends on task characteristics and computational constraints. Threshold-based selection adapts the tool set size to task complexity, while top-k selection provides predictable resource utilization. This RAG-based filtering mechanism ensures that the specialized agent operates with a focused, relevant tool set for each specific task, thereby improving both efficiency and performance.

## 2.5 Specialized Agent Architecture

The final specialized agent $\pi_{\text{specialized}}$ integrates the master agent's core reasoning capabilities together with the curated MCP box and RAG-based tool selection mechanism. The agent architecture comprises:

- **Task Analyzer**: Processes incoming tasks and generates appropriate embedding representations
- **MCP Retriever**: Implements the RAG-based selection algorithm to identify relevant tools
- **MCP Executor**: Provides runtime support for dynamic tool invocation with standardized interfaces

The inference process follows a structured pipeline that accommodates both selection strategies. A detailed workflow is shown in Algorithm algorithm 1.

Through this systematic approach, ALITA-G automatically constructs specialized agents that inherit the master agent's reasoning capabilities while being equipped with task-specific, efficiently retrievable tools, thereby achieving superior performance on target task domains with minimal manual intervention.

## 3 Experiments

Through extensive experiments on diverse task domains, we demonstrate that ALITA-G produces automatically generated agents that consistently surpass general-purpose agents in both accuracy and efficiency.

### 3.1 Experimental Setup

**Settings.** Throughout all experiments, we employ a unified agent architecture consisting of a Manager Agent and a Web Agent, following the Alita framework [4]. The Manager Agent utilizes Claude-Sonnet-4 as the base model for high-level task coordination and reasoning, while the Web Agent leverages GPT-4.1 for external information retrieval and web interactions. We select the currently most powerful text embedding model, OpenAI's text-embedding-3-large [18], as the embedding computation model, and employ threshold mode for filtering, incorporating MCPs with similarity scores greater than $\tau = 0.7$ for usage. We use GAIA [14], PathVQA [15] and The Humanity's Last Exam (HLE) [16] as benchmarks, details can be found in Appendix C. We report both the accuracy achieved on these benchmarks and the average number of tokens consumed during answer generation.

**Baselines.** We compare our approach against several state-of-the-art agent systems and variants of our method:

- **Octotools** [19]: A tool-augmented agent framework that provides agents with access to a predefined collection of specialized tools for various tasks.
- **ODR-smolagents** [20]: The Open Deep Research agent implementation within the Smolagents framework, representing a strong baseline for general-purpose agent capabilities.
- **Original Agent System**: The master agent used for MCP generation, evaluated without access to the specialized MCP box to establish the baseline performance of the underlying architecture.

| Method | Metric | GAIA | | | | PathVQA | HLE |
|--------|--------|---------|---------|---------|-------|---------|-----|
| | | Level 1 | Level 2 | Level 3 | Total | | |
| *Baseline Methods* | | | | | | | |
| Octotools | Accuracy (%) | - | - | - | 18.04 | 47 | - |
| | Avg. Tokens | - | - | - | - | - | - |
| ODR-smolagents | Accuracy (%) | 67.92 | 53.49 | 34.62 | 55.15 | 42 | - |
| | Avg. Tokens | - | - | - | - | - | - |
| *Original Agent System* | | | | | | | |
| Original (pass@1) | Accuracy (%) | 77.36 | 76.74 | 65.38 | 75.15 | 52 | 24 |
| | Avg. Tokens | 11058 | 12467 | 14308 | 12305 | 12542 | 14730 |
| Original (pass@3) | Accuracy (%) | 88.68 | 89.53 | 76.92 | 87.27 | 63 | 39 |
| | Avg. Tokens | 10947 | 12492 | 14489 | 12310 | 12627 | 14503 |
| *Generated Agents (Our Method)* | | | | | | | |
| ALITA-G$^{1\times}$ (pass@1) | Accuracy (%) | 84.91 | 80.23 | 69.23 | 80.00 | 56 | 28 |
| | Avg. Tokens | 10149 | 11357 | 13094 | 11243 | 10867 | 13128 |
| ALITA-G$^{1\times}$ (pass@3) | Accuracy (%) | 90.56 | 89.53 | 80.77 | 88.48 | 64 | 41 |
| | Avg. Tokens | 10259 | 11297 | 13027 | 11236 | 10862 | 13096 |
| ALITA-G$^{3\times}$ (pass@1) | Accuracy (%) | 86.80 | 83.72 | 73.08 | 83.03 | 60 | 33 |
| | Avg. Tokens | 9951 | 10258 | 11746 | 10394 | 10574 | 11956 |
| ALITA-G$^{3\times}$ (pass@3) | Accuracy (%) | **90.56** | **90.70** | **80.77** | **89.09** | **66** | **42** |
| | Avg. Tokens | 10025 | 10367 | 11689 | 10465 | 10479 | 12002 |

Table 1: Performance comparison across benchmarks and baseline methods. Each method is evaluated on both test accuracy and computational efficiency (measured by average token consumption). **Original** refers to the master agent system used to generate MCP boxes for specialized agents. ALITA-G$^{1\times}$ and ALITA-G$^{3\times}$ represent our method equipped with MCP boxes generated from single and triple task executions respectively. pass@1 and pass@3 indicate single-attempt and best-of-three-attempts evaluation protocols. Bold values indicate the best performance in each category.

## 3.2 EXPERIMENTAL RESULTS

Table 1 presents the comprehensive evaluation results across all benchmarks and baseline configurations.

Our experimental results demonstrate several key findings that validate the effectiveness of the proposed ALITA-G framework:

**Superior Task-Specific Performance.** The automatically generated agents consistently outperform both general-purpose baselines and the original agent system across all benchmarks. ALITA-G (3×) pass@1 achieves 83.03% accuracy on GAIA, representing a 50.5% relative improvement over ODR-smolagents (55.15%) and a 10.3% improvement over the original agent system with pass@1 (75.15%). Similar performance gains between ALITA-G (3×) pass@1 and original agent system pass@1 are observed on PathVQA (60% vs. 52%) and HLE (33% vs. 24% ), demonstrating the generalizability of our approach across diverse task domains.

**MCP Box Quality Correlation.** The comparison between single-generation and triple-generation MCP boxes reveals a clear correlation between MCP box richness and agent performance. The triple-generation variant consistently achieves higher accuracy across all benchmarks, with notable improvements on GAIA (83.03% vs. 80.00%) and more substantial gains on complex reasoning tasks in PathVQA and HLE. This finding supports our hypothesis that multiple execution rounds lead to more comprehensive and robust tool collections.

**Computational Efficiency Gains.** Remarkably, our specialized agents achieve superior accuracy while demonstrating significantly improved computational efficiency. ALITA-G (3×) reduces average token consumption to 10,394 on GAIA compared to 12,305 for the original baseline, representing a 15.5% efficiency improvement. This dual benefit of enhanced performance and reduced computational cost stems from the targeted nature of the MCP box, which provides agents with precisely the tools needed for specific task categories, eliminating extensive tool search processes.

| Iter. | Level 1 | Level 2 | Level 3 | Average | # MCPs | # Clusters | Mean Sim. | Median Sim. |
|-------|---------|---------|---------|---------|--------|------------|-----------|-------------|
| 1 | 84.91 | 80.23 | 69.23 | 80.00 | 26 | 26 | 0.28 | 0.27 |
| 2 | 84.91 | 81.40 | 71.15 | 81.82 | 46 | 41 | 0.31 | 0.29 |
| 3 | 86.79 | 82.56 | 73.08 | 83.03 | 74 | 52 | 0.30 | 0.28 |
| 4 | 86.79 | 82.56 | 73.08 | 83.03 | 102 | 60 | 0.32 | 0.30 |
| 5 | 86.79 | 83.72 | 73.08 | 83.63 | 128 | 65 | 0.34 | 0.31 |

Table 2: Performance and MCP Box statistics versus the number of generation iterations $k$ on the GAIA validation set. Accuracies are reported in % for each difficulty level and their average. **# MCPs** is the count of curated MCPs in the MCP Box after filtering and abstraction. **Mean/Median Sim.** mean statistics of pairwise cosine similarity between MCP embeddings, where Mean Sim. denotes the average pairwise similarity, Median Sim. denotes the Median of pairwise similarities. **# Clusters** is the number of connected components when linking MCP pairs with similarity ($\geq 0.7$, serving as a proxy for the number of independent MCPs. **Iter.** denotes how many times the original task set is run when constructing the MCP Box.

The consistent improvements across multiple evaluation dimensions provide strong empirical evidence for the effectiveness of our automatic agent generation methodology. These results demonstrate that task-driven MCP curation, combined with intelligent retrieval mechanisms, enables the creation of specialized agents that surpass general-purpose systems in both performance and computational efficiency.

# 4 ANALYSIS

## 4.1 ANALYSIS OF MCP BOX SCALABILITY

To understand the performance boundaries of MCP Box expansion and identify the optimal number of generation iterations, we investigate the relationship between MCP generation frequency and agent performance improvements across different task complexities.

**Settings.** We use the full GAIA validation set and vary the number of *generation iterations* $k \in \{1, 2, 3, 4, 5\}$. Each iteration runs the master agent once over the entire validation set to harvest additional MCPs, followed by filtering and abstraction to construct the accumulated MCP Box. For each $k$ we report: (i) the number of curated MCPs; (ii) summary statistics (mean and median) of pairwise MCP similarity; and (iii) the number of clusters under a fixed similarity threshold. Concretely, we embed each MCP by concatenating its *description* and *use-case* fields, encoding the resulting text with `text-embedding-3-large`, and $\ell_2$-normalizing the embedding. Cosine similarity between two MCPs is then the inner product of their normalized embeddings. To quantify redundancy, we build an undirected similarity graph whose vertices are MCPs and whose edges connect pairs with similarity at least $\tau = 0.7$; the reported cluster count is the number of connected components in this graph. Downstream agent performance is evaluated with the accumulated MCP Box while keeping all other configurations identical to Section 3.

**Results. Performance shows substantial gains from iterations 1 to 3 before exhibiting clear saturation and diminishing returns.** Table 2 exhibits a clear pattern of diminishing returns in MCP Box scalability. The largest gains occur when increasing the number of generations from $k=1$ to $k=3$, with average accuracy rising from $80.00\%$ to $83.03\%$. We attribute this improvement to the stochasticity of MCP discovery: the master agent does not consistently surface the most useful MCPs in a single pass, and multiple passes enrich coverage of the task distribution. Beyond $k=3$, additional iterations yield marginal benefits—the average remains flat at $k=4$ and nudges to $83.63\%$ at $k=5$. Per-level trends echo this picture: Level 1 saturates by $k=3$ ($84.91{\rightarrow}86.79$), Level 3 plateaus thereafter ($69.23{\rightarrow}73.08$), and the modest late-stage gain is concentrated in Level 2 ($82.56$ at $k=3$ to $83.72$ at $k=5$)

**Similarity analysis reveals progressive redundancy accumulation that explains the performance plateau.** Complementing these performance trends, the similarity and clustering statistics indicate increasing redundancy as the MCP Box grows. Under the fixed threshold $\tau{=}0.7$, the number of

| Metric | 1 Generation | 2 Generation | 3 Generation |
|---|---|---|---|
| Overall accuracy (%) | 80.00 | 81.52 | 83.03 |
| Wrong→Right # (vs. baseline) | 9 | 12 | 13 |
| Right→Wrong # (vs. baseline) | 1 | 1 | 0 |
| Avg. MCP calls per question | 1.9 | 2.2 | 2.4 |
| Avg. MCP calls per improved questions | 2.7 | 3.0 | 3.4 |

Table 3: MCP usage and outcome metrics on the GAIA validation set across MCP Box configurations. 1 Generation, 2 Generation, and 3 Generation refer to MCP Boxes constructed via one, two, and three iterative generation rounds. **Wrong→Right # (vs. baseline)** counts items that the baseline agent (without an MCP Box) answers incorrectly, but the integrated agent answers correctly. **Right→Wrong # (vs. baseline)** counts items that the baseline answers correctly, but the integrated agent answers incorrectly. **Avg. MCP calls per question** is the mean number of calls to any MCP per question over all instances. **Avg. MCP calls per improved question** are the same mean computed only over the Wrong→Right subset.

connected components—our proxy for effective MCP families—increases sublinearly relative to the total number of curated MCPs: clusters grow from 26 to 65 while MCPs grow from 26 to 128. Consequently, the effective-coverage ratio (#Clusters/#MCPs) drops from $1.00$ ($k=1$) to $0.51$ ($k=5$), and the marginal yield of new, independent clusters per iteration diminishes (+15, +11, +8, +5 from $k=1{\rightarrow}5$). The performance plateau between $k=3$ and $k=4$ coincides with an addition of 28 MCPs but only 8 new clusters alongside a rise in average similarity, suggesting that later iterations predominantly introduce near-duplicates or narrow variants of existing capabilities. Taken together, these results indicate that $k=3$ offers a favorable balance between computational cost and utility—capturing most of the diverse, high-impact MCP families while avoiding the redundancy that characterizes further expansions.

## 4.2 MCP BEHAVIOR ANALYSIS

To validate that agents indeed gain enhanced capabilities through MCP Box integration, we conduct a detailed analysis of MCP usage patterns in generated agents.

**Settings.** We analyze MCP usage behavior on the GAIA validation set using agents equipped with MCP Boxes generated through 1, 2, and 3 iterative rounds, donated as **1/2/3 Generation**. Beyond usage metrics, we additionally report (a) *overall accuracy* after MCP Box integration, (b) the number of questions flipped from wrong to right (*Wrong→Right*) relative to the baseline (no MCP Box), and (c) the number flipped from right to wrong (*Right→Wrong*). All metrics are computed on the same GAIA validation set. We continue to track the average number of MCP calls per question over all instances and specifically for *improved questions* (incorrect under the baseline but correct after integration). An *MCP call* refers to one invocation to any MCP in the connected MCP Box; the same MCP may be called multiple times within a single task.

**Results.** Table 3 shows a clear trend of increased MCP utilization as the MCP Box becomes more mature. The average number of MCP calls per question rises monotonically from 1.9 to 2.4 when moving from 1 to 3 generations, while the corresponding average on improved questions increases from 2.7 to 3.4. Notably, improved questions consistently elicit substantially more MCP usage than the overall average—about $1.4\times$ more in all configurations (2.7/1.9= 1.42, 3.0/2.2= 1.36, 3.4/2.4= 1.42). The marginal increments suggest targeted deployment of MCPs on challenging instances: overall usage grows by +0.3 then +0.2 calls, whereas improved-question usage grows by +0.3 then +0.4, indicating that later generations concentrate additional tool use where it is most impactful.

Turning to answer correctness, overall accuracy improves steadily from 80.00% to 83.03% as the number of generations increases from 1 to 3 (a gain of +3.03 points). These gains are driven primarily by Wrong→Right flips (9/12/13), while Right→Wrong flips are rare (1/1/0), yielding net improvements of +8, +11, and +13 respectively. The low incidence of regressions—vanishing

by the 3-generation setting—indicates that the method is robust: it rarely converts correct baseline answers into errors while delivering consistent accuracy gains as the MCP Box is strengthened. Upon closer examination of the Right→Wrong cases in the 1- and 2-generation settings, we observe that these involve distinct questions and stem from reasoning errors introduced during agent execution rather than incorrect MCP usage. These regressions appear attributable to inherent LLM robustness limitations within the agent system rather than deficiencies introduced by MCP Box integration.

## 5 RELATED WORKS

### 5.1 AUTO GENERATING AGENT

Recent advances in automated agent construction have focused on generating agents or agent systems with varying degrees of automation and scope. AutoAgents [21] pioneers automatic multi-agent generation by dynamically creating specialized agents. Building on this foundation, AutoGenesis-Agent [22] introduces self-generating capabilities with lifecycle management for multi-agent systems, while EvoAgent [23] applies evolutionary algorithms to extend expert agents into multi-agent configurations. MetaGPT [24] incorporates human software development workflows into LLM-based multi-agent collaboration. More recently, AutoAgent [25] provides a zero-code framework for creating LLM agents, and Dynamic LLM-Agent Network [26] focuses on automatic agent team optimization without requiring strong human priors. Our work differs fundamentally by generating complete, task-specific agents ready for downstream deployment, rather than focusing on isolated component generation or requiring extensive manual configuration for integration.

### 5.2 SELF-EVOLVING AGENT

Self-evolving agents represent a paradigm where AI systems autonomously improve their capabilities through iterative learning and adaptation. Recent comprehensive surveys [2] categorize these systems based on their evolution mechanisms, ranging from parametric updates to non-parametric component optimization. Early foundational work includes Reflexion [27], which introduces verbal reinforcement learning for language agents through self-reflection and memory-based learning, and ExpeL [28], which enables agents to gather and learn from experiential data across training tasks autonomously. More recent advances have explored diverse self-evolution mechanisms such as SAGE [29], Agent-Pro [30], Gödel Agent [31], RAGEN [32], EvolveSearch [33] and SELF [34]. Our framework can be conceptualized as a form of agent self-evolution, where agents leverage previously generated tools from past task executions to enhance performance on similar future tasks, achieving both improved accuracy and computational efficiency.

### 5.3 MCP

Model Context Protocol (MCP) has emerged as a standardized framework for enabling seamless integration between AI systems and external tools or data sources [35; 36; 37; 38]. Introduced by Anthropic [13], MCP provides a unified interface that addresses fragmentation challenges in tool integration for LLM-based agents. Our methodology relies on constructing high-quality MCP boxes as the foundation for generating specialized agents, where the richness and relevance of the MCP collection directly correlates with the resulting agent's task-specific performance. While [39] also leverages the MCP as a conduit for distilling capabilities across agents, their focus is on curating strong teacher agents to assist weaker ones. In contrast, our work targets the end-to-end evolution of a more powerful domain-specialist agent tailored to a specific target domain, moving beyond assistance to specialization.

## 6 CONCLUSION

In this paper, we introduce ALITA-G, a novel self-evolution framework that transforms generalist agents to domain-specific experts. By organizing task-derived tools into MCP Boxes with RAG, our approach significantly enhances agent capabilities on specific domain tasks. Future work could further expand the ways agents perform self-evolution, enabling even greater leaps in agent development through collaborative enhancement across multiple dimensions beyond the current framework.

## 7 REPRODUCIBILITY STATEMENT

We will open-source our code and evaluation scripts upon publication. All datasets, model settings are described in the paper, enabling researchers to reproduce all reported experiments and results.

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

## A   THE USE OF LLMS

LLMs did not play an important role in this paper's research ideation or writing to the extent that they should be regarded as a contributor. In the experiments, LLMs are the main experimental object.

## B   ALGORITHM DETAILS

---

**Algorithm 1:** Specialized Agent Inference

---

1: **Input:** Task query $x_{\text{new}}$, MCP box $\mathcal{B}$, selection mode mode $\in \{\text{threshold}, \text{top-k}\}$, parameter $\theta$ (threshold $\tau$ or $k$)
2: $\mathbf{e}_{\text{query}} \leftarrow \phi(x_{\text{new}})$
3: **for** $m = 1$ to $M$ **do**
4:     $\mathbf{e}_m \leftarrow \phi(\text{description}_m \oplus \text{use\_case}_m)$
5:     $s_m \leftarrow \text{cosine\_similarity}(\mathbf{e}_{\text{query}}, \mathbf{e}_m)$
6: **end for**
7: **if** mode $=$ threshold **then**
8:     $\mathcal{B}_{\text{filtered}} \leftarrow \{\widehat{\text{MCP}}_m \mid s_m \geq \theta, m \in [M]\}$
9: **else if** mode $=$ top-k **then**
10:     $\mathcal{B}_{\text{filtered}} \leftarrow \text{Top-k-Select}(\{s_m\}, \mathcal{B}, \theta)$
11: **end if**
12: context $\leftarrow$ Initialize($x_{\text{new}}, \mathcal{B}_{\text{filtered}}$)
13: **while** not task_completed **do**
14:     reasoning_step $\leftarrow$ ReasoningEngine(context)
15:     **if** tool_required **then**
16:         mcp $\leftarrow$ SelectTool($\mathcal{B}_{\text{filtered}}$)
17:         result $\leftarrow$ MCPExecutor(mcp, args)
18:         context $\leftarrow$ Update(context, result)
19:     **end if**
20: **end while**
21: **Return:** Final output $y_{\text{predicted}}$

---

## C   BENCHMARK DETAILS

We evaluate our framework on three challenging benchmarks that span different domains and complexity levels:

- **GAIA** [14]: The General AI Assistant (GAIA) is a benchmark that comprises 466 real-world questions across three difficulty levels, testing agents' capabilities in web browsing, tool usage, and complex reasoning. The benchmark includes questions ranging from simple factual queries that require only single-tool usage to multi-step reasoning tasks that necessitate extensive tool coordination. We use the complete validation set.

- **PathVQA** [15]: PathVQA is a medical visual question answering benchmark containing pathology images paired with questions. The dataset requires specialized domain knowledge and visual reasoning capabilities. Due to resource constraints, we randomly sample 100 representative examples for evaluation.

- **HLE** [16]: The Humanity's Last Exam (HLE) is a challenging academic benchmark that focuses on complex reasoning tasks that require multi-modal understanding and sophisticated problem-solving strategies. Similar to PathVQA, we sample 100 examples to balance comprehensive evaluation with computational efficiency.

| Method | Level 1 | Level 2 | Level 3 | Average |
|---|---|---|---|---|
| RAG with Description+Use Case | **86.80** | **82.55** | **73.08** | **83.03** |
| RAG with Description | 84.91 | 81.39 | 73.08 | 81.82 |
| RAG with Use Case | 83.01 | 79.06 | 61.53 | 77.57 |

Table 4: Performance comparison of different RAG content configurations on GAIA validation set using triple-generation MCP boxes. **RAG with Description** refers to searching by the description of the MCP function, while **RAG with Use Case** refers to searching by the task when generating this MCP, and **RAG with Description+Use Case** refers to searching by combining the two.

| Strategy | Parameters and Accuracy (%) | | | | | |
|---|---|---|---|---|---|---|
| Threshold ($\tau$) | 0.65 | 0.70 | 0.75 | 0.80 | 0.85 | 0.90 |
| Accuracy | 76.0 | **84.0** | 80.0 | 76.0 | 76.0 | 68.0 |
| Top-k ($k$) | 1 | 2 | 3 | 5 | 10 | 20 |
| Accuracy | 76.0 | 80.0 | 80.0 | 76.0 | 76.0 | 72.0 |

Table 5: Performance comparison of different MCP selection strategies on GAIA validation subset. **Threshold-based selection** filters MCPs by semantic similarity scores above threshold $\tau$, while **Top-k selection** retrieves the $k$ most similar MCPs regardless of absolute similarity values. Results show that threshold-based selection with $\tau = 0.70$ achieves optimal performance.

## D    ADDITIONAL ANALYSIS

### D.1    ANALYSIS OF RAG CONTENT COMPONENTS

To understand the contribution of different components in our RAG-based MCP selection mechanism, we evaluate the impact of using different textual representations for computing semantic embeddings.

**Settings.**    We test three configurations: using only MCP descriptions for RAG, using only the use cases that triggered MCP generation for RAG, and using the concatenation of both description and use case (our main experimental setting). We compare agent performance under these settings on the GAIA validation set, with all other experimental configurations kept consistent with the main experiments section 3.

**Results.**    The results are presented in  Table 4.  The results demonstrate that combining both description and use case information achieves the best performance across all difficulty levels, with an average accuracy of 83.03%. Using description alone for RAG achieves competitive performance (81.82%), while using only use case information results in notably lower performance (77.57%). This indicates that MCP descriptions provide more generalizable semantic information for tool selection, while use case information, though valuable when combined with descriptions, is less effective as a standalone retrieval signal.

### D.2    ANALYSIS OF MCP SELECTION STRATEGIES

To understand the impact of different MCP selection mechanisms on agent performance, we evaluate various MCP filtering approaches during the task execution phase, including threshold-based selection, top-k selection, and different filtering thresholds.

**Settings.**    We experiment with threshold values $\tau \in \{0.65, 0.70, 0.75, 0.80, 0.85, 0.90\}$ and top-k values $k \in \{1, 2, 3, 5, 10, 20\}$. We sample 25 questions from the GAIA Validation Set for testing (9 Level 1, 12 Level 2, and 4 Level 3 questions, maintaining the distribution of the validation set across

| Embedding Encoder | Accuracy (%) |
|---|---|
| text-embedding-3-large | **84.0** |
| text-embedding-3-small | 80.0 |
| Qwen3-Embedding-8B | 76.0 |
| NV-Embed-v2 | 72.0 |
| BGE-M3 | 72.0 |

Table 6: Performance comparison of different embedding encoders of RAG. The **text-embedding-3-large** and **text-embedding-3-small** refers to OpenAI's corresponding embedding model.

the three levels). The experiments use the MCP Box generated through triple executions on GAIA validation, with all other settings kept consistent with the main experiments section 3.

**Results**  The results are presented in Table 5. The results demonstrate that threshold-based selection generally outperforms top-k selection. This may be attributed to the fact that different tasks require varying numbers of MCPs from the MCP Box. Fixed top-k selection cannot adapt well to all tasks—some tasks cannot utilize all suitable MCPs, while others receive irrelevant MCPs. When using threshold-based selection, both excessively high and low thresholds harm performance. This is understandable: low thresholds select task-irrelevant MCPs, while high thresholds exclude useful MCPs that should be selected.

### D.3 ANALYSIS OF EMBEDDING ENCODERS

We evaluate the impact of different embedding encoders on the RAG-based MCP selection mechanism. The choice of encoder directly affects the quality of semantic similarity computation, which is crucial for retrieving relevant MCPs during task execution.

**Settings.**  We compare several state-of-the-art embedding models, including proprietary models OpenAI's text-embedding-3-large [18], text-embedding-3-small [18], and open-source models Qwen3-Embedding-8B [40], NV-Embed-v2 [41], and BGE-M3 [42]. We use the same 25 questions sampled from GAIA validation as in subsection D.2. All other experimental settings remain consistent with the main experiments section 3.

**Results.**  The results are presented in Table 6. The results demonstrate that high-quality encoders significantly impact task performance. More capable encoders help the model identify suitable MCPs more effectively, thereby enabling greater improvements in task-solving capabilities. This finding highlights the importance of encoder selection in retrieval-augmented agent architectures.

## E  CASE STUDY

We visualize the core mechanism of ALITA-G: task-driven MCP creation, its abstraction into a reusable primitive, and the downstream effect on inference. Figure 2 illustrates how a raw, task-bound MCP (left) produced during a marine biology literature task is abstracted into a parameterized, FastMCP-compatible tool with standardized interfaces and documentation (right). This abstraction converts ephemeral, instance-specific solutions into broadly reusable capabilities that can be reliably retrieved across tasks.

Figure 3 demonstrates the impact at inference time. For a thermodynamics question, the baseline agent without an MCP Box fails (predicting 20 mL), whereas the specialized agent retrieves the abstracted `extract_pdf_measurement` via MCP-level RAG and solves the problem correctly (55 mL). The comparison highlights that (i) abstraction is crucial for turning ad-hoc tool creations into general-purpose components, and (ii) the MCP Box materially improves accuracy by enabling targeted, retrieval-augmented tool selection at run time.

---

**Initial Task**

What integer-rounded percentage of the total length of the harlequin shrimp recorded in Omar Valencia-Mendez 2017 paper was the sea star fed to the same type of shrimp in G. Curt Fiedler's 2002 paper?

**Raw MCP**

```
def download_and_extract_pdf():
        url = "<some specific url>"
        …
    with open(filename, 'wb') as f:
            f.write(response.content)
    text = extract_text(filename)
    return text

def extract_measurements():
        patterns = [ r'(\d+\.?\d*)\s…]
```

**Abstracted MCP**

```
def extract_pdf_measurement(pdf_url, target_units=None,
search_terms=None):
        """
        Extract measurements from scientific PDFs with flexible unit
        specification
        Args:
        pdf_url (str): URL of the PDF to download and process
        target_units (list): Specific units to search for (e.g., ['ml', 'L',
        'cm³'])
        search_terms (list): Context terms to focus search
        """
```

Figure 2: **MCP generation and abstraction.** *Left:* A raw MCP emerges during execution to extract measurements from scientific PDFs in response to a concrete task. *Right:* The MCP is abstracted, where hard-coded values are lifted into parameters, interfaces are standardized to FastMCP, and documentation is enhanced, yielding a reusable tool suitable for retrieval and reuse across tasks.

**New Task**

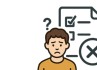

What is the volume in milliliters of a system comprised of 0.312 kg Freon-12 refrigerant when placed at the bottom of the Marianas Trench and allowed to stabilize at the Trench's peak temperature, rounded to the nearest mL? Provide your answer as just an integer value.

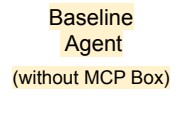

**Baseline Agent**

(without MCP Box)

**Data Collection Attempts:**
    Attempted to find Freon-12 thermodynamic properties: Failed to obtain accurate data
    …
**Calculation with Incomplete Data:**
    Used approximate values and simplified assumptions.

Incorrect answer

(20 mL) ❌

**Specialized Agent**

(with MCP Box)

**Call relevant MCP:**
    extract_pdf_measurement(
    pdf_url="https://doi.org/10.1016/j.fluid.2018.03.021",
    target_units=['ml', 'L', 'cm³', 'kg/m³'],
    search_terms=['Freon-12', 'density', 'volume', 'pressure', 'temperature']
    )

Correct answer

(55 mL) ✓

Figure 3: **Effect of the MCP Box at inference.** *Baseline agent (no MCP Box):* fails to obtain precise thermodynamic properties and answers incorrectly (20 mL). *Specialized agent (with MCP Box):* retrieves the abstracted extract_pdf_measurement via RAG, extracts the needed properties, and answers correctly (55 mL). The example underscores how abstraction plus MCP-level retrieval converts transient problem-solving into reusable competence that boosts downstream performance.

