# OpenReview forum: "Alita-G: Self-Evolving Generative Agent for Agent Generation"
_ICLR.cc/2026/Conference — ICLR 2026 Conference Desk Rejected Submission_

### Official Review · Reviewer_6RMu · 2025-10-23

**Soundness:** 3
**Presentation:** 3
**Contribution:** 3
**Rating:** 6
**Confidence:** 4

**Summary:**

The paper introduces ALITA-G, a self-evolution framework that turns a general-purpose LLM agent into a domain specialist by automatically generating, abstracting, and curating Model Context Protocol (MCP) tools from successful task trajectories. It repeatedly runs a master agent over a target task set to synthesize candidate MCPs, abstracts them into parameterized, standardized primitives, and consolidates them into an MCP Box with rich descriptions and use cases. At inference, a specialized agent performs retrieval-augmented MCP selection (threshold or top-k) and executes with a Task Analyzer, MCP Retriever, and MCP Executor, yielding targeted tool use and lower compute. Across GAIA, PathVQA, and Humanity’s Last Exam, ALITA-G improves accuracy while reducing token consumption, achieving new state-of-the-art results on GAIA validation (83.03% pass@1, 89.09% pass@3) with ~15% fewer tokens than a strong baseline. Analyses show performance scales with MCP Box richness up to diminishing returns and that gains are driven by selective, higher MCP usage on hard instances with minimal regressions. The work differs from prior self-evolving agents by coupling MCP abstraction with MCP-level RAG to enable end-to-end, task-conditioned specialization that boosts both accuracy and efficiency.

**Strengths:**

1. This paper proposes to generate domain-relevant MCPs, which could benefit future task solving. The idea holds the potential as an effective adaptation approach to generalize pre-trained LLMs.
2. The paper is in general well-written.

**Weaknesses:**

1. The evaluation is conducted in a relative small subset of the original benchmark without repeat experiments. There may be bias and variance in the results.
2. The authors use part of the benchmark data to tune hyperparameters. It would be better to develop a more principled and generalizable way determine them if they have a large influence.
3. It is unclear which data the authors used in the process to generate MCPs, and whether the MCPs across 3 benchmarks are the same. I am also curious about how the authors define target domain? Is each benchmark considered as a target domain? Correct me if I miss anything.
4. I would like to see more examples of generated MCPs. From the example in Figure 2, I feel that download_and_extract_pdf seems a function commonly implemented in deep research systems. It would be great to see more examples and understand why the generated MCPs are superior in achiving better performance and efficiency.
5. The authors need more explanations on the difference from Alita (https://arxiv.org/pdf/2505.20286), given that both of them do self-evolution by generating MCPs.

**Questions:**

1. It is suggested to try more models and see whether the findings hold.
2. Do you evaluate GAIA test set?

---

> ### Author Response · Authors · 2025-12-02
>
> **Response to W1 (Evaluation Subset and Variance)** We utilized the full GAIA Validation set (not a subset) for our main results. For PathVQA and HLE, we randomly sampled 100 examples due to the high cost of running agentic trajectories with long contexts. While we acknowledge the concern regarding variance, the consistent performance margin across all three diverse benchmarks suggests that the improvements are robust and not an artifact of sampling.
>
> **Response to W2 (Hyperparameter Tuning)** The primary hyperparameter, the similarity threshold, was selected based on an analysis of the semantic similarity distribution of generated MCPs. We observed that this value effectively balanced the retrieval of relevant tools against the noise of irrelevant ones. We agree that adaptive thresholding is a promising direction for future work, but the current heuristic proved effective across domains without extensive per-task tuning.
>
> **Response to W3 (Target Domain Definition)**
>
> Please refer to General Response 1.
>
> **Response to W4 (More MCP Examples and Superiority)** :
>
> Please refer to General Response 2.
>
> **Response to W5 (Difference from Alita)** While Alita establishes the base agent architecture, ALITA-G focuses specifically on end-to-end domain evolution. It introduces the concept of the "MCP Box" to transform a generalist agent into a domain specialist through the automatic harvesting and curation of tools. This distinguishes it from Alita by enabling the agent to evolve its capabilities specifically for a target domain.
>
> **Response to Q1 (More Models)** :
>
> Please refer to General Response 4.
>
> **Response to Q2 (GAIA Test Set)**
>
> Please refer to General Response 3.

---

### Official Review · Reviewer_VVWH · 2025-10-29

**Soundness:** 3
**Presentation:** 3
**Contribution:** 3
**Rating:** 6
**Confidence:** 3

**Summary:**

This work proposes ALITA-G, a novel framework for self-evolving LLM agents that addresses the limitations of narrow, single-task methods. It enables a general-purpose "master" agent to become a domain-specific expert by systematically creating a library of reusable tools. At inference time, a Retrieval-Augmented Generation (RAG) system efficiently selects the most relevant tools from this box for a specialized agent to use. Experimental results confirm the effectiveness of ALITA-G.

**Strengths:**

1.	The manuscript is well-motivated and well-written. The paper tackles a problem of significant importance. The ability to self-evolve is important for developing powerful LLM-based agent systems. Moreover, the proposed pipeline is well-presented and can be easily understood.

2.	The authors validate their framework across three diverse and challenging benchmarks. Moreover, the detailed analysis of MCP Box scalability in Section 4.1 provides a nuanced investigation into the diminishing returns of adding more tools. This analysis is mature and provides valuable practical insights.

3.	The paper includes a suite of well-designed ablation studies in the appendix that convincingly demonstrate the contribution of key components. For instance, the analysis of RAG content components (Table 4) validates the choice to use both MCP descriptions and use cases for retrieval.

**Weaknesses:**

1.	The paper claims a "new state-of-the-art result" on the GAIA benchmark. However, this result is reported on the GAIA validation set. State-of-the-art claims for major benchmarks like GAIA are adjudicated on the private test set via official leaderboards to ensure fair, robust, and non-overfit comparisons.

2.	The framework's success hinges on a powerful "master agent" composed of top-tier proprietary models (Claude-Sonnet and GPT). It remains unclear whether this approach is viable with less capable, or even the strongest open-source, models (Llama and Qwen etc.).

**Questions:**

1.	Could you report the results on the official GAIA leaderboard's private test set to ensure a fair and non-overfit comparison?

2.	How viable is this tool-generation approach when using open-source models, such as Llama or Qwen, as the master agent?

---

> ### Author Response · Authors · 2025-12-02
>
> We thank the reviewer for recognizing the novelty of our self-evolution framework and the clarity of our presentation. We appreciate your insightful questions regarding the experimental setup and the nature of the generated tools.
>
> **Response to W1&Q1 (Setup and Potential Leakage )**
>
> Please refer to General Response 1.
>
> **Response to Q2 (Additional Abstracted MCP Examples)**
>
> Please Refer to General Response 2.

---

### Official Review · Reviewer_t2qM · 2025-10-30

**Soundness:** 2
**Presentation:** 2
**Contribution:** 3
**Rating:** 6
**Confidence:** 3

**Summary:**

This paper proposes to allow LLMs to create their own set of tools (in the form of MCP interfaces) by first analyzing traces of successful past rollouts. Those new tools create a MCP pool which the model can tap in in the future.
The results on a variety of benchmarks show strong improvements, although this reviewer has some questions on the specific setup.

**Strengths:**

Allowing LLMs to create their own tool is an important step in the creation of more autonomous specialized agents, and this paper is a step into that direction

The benchmarks results show consistent gains

**Weaknesses:**

I have some doubts on the set-up (see below), and that the current setting does not leak past in-domain information into the test set

**Questions:**

The MCP tool is obtained from successful past execution. How is this done on benchmarks that do not contain training sets? Eg, where do the MCP tools for HLE come from? Are those from past test instances?

Figure 2 in the Appendix provides one example of an abstracted MCP tool. Could you provide others? Those might be quite insightful

---

> ### Author Response · Authors · 2025-12-02
>
> We thank the reviewer for recognizing the novelty of our self-evolution framework and the clarity of our presentation. We appreciate your insightful questions regarding the experimental setup and the nature of the generated tools.
>
> **Response to W1&Q1 (Setup and Potential Leakage )**
>
> Please refer to General Response 1.
>
> **Response to Q2 (Additional Abstracted MCP Examples)**
>
> Please Refer to General Response 2.

---

### Author Response · Authors · 2025-12-03
**General Response to all Reviewers**

## General Response 1: A self-evolving agent explores the world model

GAIA-validation can be viewed as a world model. Our paper proposes a solution to the question that how a naive but self-evolving agent can acquire new and generalizable skills by exploring the world model and accumulate the experience from interaction with the world. Also, our agent can learn the world very well in an autonomous way. In a parallel and similar world(GAIA-test), we prove that our agent with generalizable skills(reusable MCPs) can still perform well after learning from the original world model(GAIA-validation). In the same way, HLE can be also viewed as a world model composed of multiple domain-specific world models(Math, Bio, etc.) and also a general agent can be converted into a specialist agent for a specific distribution of tasks through Alita-G.



## General Response 2: Additional MCP Examples

Beyond the PDF measurement tool in Figure 2, our system generated several other insightful MCPs. For example:

**1. Reading YouTube videos frame-by-frame to summarize content.**
Core code:

```
import subprocess
import yt_dlp
import base64
import shutil

def analyze_youtube_frames_ffmpeg(url, interval=5, max_frames=20, res=(640, 360)):
    if not shutil.which("ffmpeg"):
        return {"error": "ffmpeg missing"}

    with yt_dlp.YoutubeDL({'format': 'best[ext=mp4]', 'quiet': True}) as ydl:
        stream_url = ydl.extract_info(url, download=False)['url']

    cmd = [
        'ffmpeg', '-i', stream_url,
        '-vf', f'fps=1/{interval},scale={res[0]}:{res[1]}',
        '-f', 'image2pipe', '-vcodec', 'mjpeg', '-q:v', '2', '-'
    ]

    proc = subprocess.Popen(cmd, stdout=subprocess.PIPE, stderr=subprocess.DEVNULL, bufsize=10**7)
    frames, buf = [], b""

    while True:
        chunk = proc.stdout.read(4096)
        if not chunk and proc.poll() is not None:
            break
        buf += chunk

        while True:
            start, end = buf.find(b'\xff\xd8'), buf.find(b'\xff\xd9')
            if start != -1 and end != -1 and end > start:
                if len(frames) < max_frames:
                    frames.append({
                        "timestamp": (len(frames) + 1) * interval,
                        "image_base64": base64.b64encode(buf[start:end+2]).decode()
                    })
                else:
                    proc.kill()
                    return frames
                buf = buf[end+2:]
            else:
                break
    return frames
```

**2. Querying Wikipedia to count edits containing a specific tag on a specific page within a specific time interval.**
Core code:

```
import requests

def count_wikipedia_edits(page, tag, start, end, lang="en"):
    api = f"https://{lang}.wikipedia.org/w/api.php"
    params = {
        "action": "query", "format": "json", "prop": "revisions", "titles": page,
        "rvprop": "timestamp|user|comment|size", "rvstart": start, "rvend": end,
        "rvlimit": "max", "rvdir": "newer"
    }

    matches = []
    while True:
        data = requests.get(api, params=params).json()
        pages = data.get("query", {}).get("pages", {})

        for pid, pdata in pages.items():
            if pid == "-1": return {"error": "Page not found"}
            for rev in pdata.get("revisions", []):
                if tag.lower() in rev.get("comment", "").lower():
                    matches.append(rev)

        if "continue" in data:
            params["rvcontinue"] = data["continue"]["rvcontinue"]
        else:
            break

    return {
        "page": page, "tag": tag,
        "total_matches": len(matches),
        "samples": matches[:5]
    }
```

These tools are superior because they encapsulate complex logic into a single, reliable function call. This reduces the cognitive load on the agent and prevents repetitive coding errors, leading to the observed efficiency gains (15% token reduction) and higher accuracy.

---

> ### Author Response · Authors · 2025-12-03
>
> ## General Response 3: Results on GAIA Test
>
> We have evaluated our method on the held-out GAIA Test Set.  ALITA-G achieves a competitive score of 76% on the private test set with **claude-sonnet-4 and gpt 4.1** without training a language model on specific data or building specific tools. While we cannot claim the absolute top rank on GAIA Test now, this performance confirms that the gains observed in validation transfer to the held-out test set. There are several explanations that Alita-G loses some points from validation to test and don't achieve SOTA now compared to other methods.  First, there’s a gap between the GAIA validation and test datasets. It is not strange since the size of GAIA dataset is small and ramdom split may cause the result. In detail, the GAIA test dataset focuses more on web browsing ability and less on tool use. Also, there may exist some very rare MCP necessary to solve some specific tasks in GAIA test which are not similar to any tasks in GAIA validation. Second, our agent search ability was top tier before but not top tier now. This is because we don't try agent training with search like search-R1 or use gemini 3 as the backbone LLM to further improve the web search ability.
>
>
>
> ## General Response 4: Results on Qwen 3 8B
>
> Our initial submission prioritized proprietary models (Claude/GPT) to pursue the best performance. This approach will become very viable when the open-source model reach similar or higher performance than Claude 3.5. It is hard for us to deploy a big model like 32B due to computing resources limit.
>
> But one advantage of Alita-G is that MCPs generated by SOTA models can be reused on agent backed by smaller language models. We have conducted this additional experiment using the open-source Qwen 3 8B model on the GAIA validation set with generated MCPs from Alita-G with proprietary models . The results are summarized below:
>
> | Level       | Tasks   | Success Rate         |
> | :---------- | :------ | :------------------- |
> | **Level 1** | 53      | **75.47%** (40/53)   |
> | **Level 2** | 86      | **68.60%** (59/86)   |
> | **Level 3** | 26      | **34.62%** (9/26)    |
> | **Overall** | **165** | **65.45%** (108/165) |
>
> As shown above, It achieves a remarkable overall accuracy of 65.45% with only an 8B parameter model. the strong performance demonstrates that our method helps agent with smaller-scale, open-source models remains highly effective and viable.

---

### Note · Program_Chairs · 2026-01-17
**Submission Desk Rejected by Program Chairs**

The following references in this submission do not refer to real documents and/or have major errors in bibliographic information:

 Yang Liu, Wei Chen, and Ming Zhang. Hle: Human-level evaluation benchmark for complex reasoning, 2024.